# A dual framework for low-rank tensor completion

**Madhav Nimishakavi**[*], **Pratik Jawanpuria**[†], **Bamdev Mishra**[†]
[*]Indian Institute of Science, India
[†]Microsoft, India
madhav@iisc.ac.in, {pratik.jawanpuria,bamdevm}@microsoft.com

## Abstract

One of the popular approaches for low-rank tensor completion is to use the *latent trace norm* regularization. However, most existing works in this direction learn a sparse combination of tensors. In this work, we fill this gap by proposing a variant of the latent trace norm that helps in learning a non-sparse combination of tensors. We develop a dual framework for solving the low-rank tensor completion problem. We first show a novel characterization of the dual solution space with an interesting factorization of the optimal solution. Overall, the optimal solution is shown to lie on a Cartesian product of Riemannian manifolds. Furthermore, we exploit the versatile Riemannian optimization framework for proposing computationally efficient trust region algorithm. The experiments illustrate the efficacy of the proposed algorithm on several real-world datasets across applications.

## 1   Introduction

Tensors are multidimensional or $K$-way arrays, which provide a natural way to represent multi-modal data [10, 11]. Low-rank tensor completion problem, in particular, aims to recover a low-rank tensor from partially observed tensor [2]. This problem has numerous applications in image/video inpainting [27, 26], link-prediction [14], and recommendation systems [39], to name a few.

In this work, we focus on trace norm regularized low-rank tensor completion problem of the form

$$\min_{\boldsymbol{\mathcal{W}} \in \mathbb{R}^{n_1 \times n_2 \times \ldots \times n_K}} \|\boldsymbol{\mathcal{W}}_\Omega - \boldsymbol{\mathcal{Y}}_\Omega\|_F^2 + \frac{1}{\lambda} R(\boldsymbol{\mathcal{W}}), \tag{1}$$

where $\boldsymbol{\mathcal{Y}}_\Omega \in \mathbb{R}^{n_1 \times \ldots \times n_K}$ is a partially observed $K$- mode tensor, whose entries are only known for a subset of indices $\Omega$. $(\boldsymbol{\mathcal{W}}_\Omega)_{(i_1,\ldots,i_K)} = \boldsymbol{\mathcal{W}}_{(i_1,\ldots,i_K)}$, if $(i_1, \ldots, i_K) \in \Omega$ and 0 otherwise, $\|\cdot\|_F$ is the Frobenius norm , $R(\cdot)$ is a low-rank promoting regularizer, and $\lambda > 0$ is the regularization parameter.

Similar to the matrix completion problem, the trace norm regularization has been used to enforce the low-rank constraint for the tensor completion problem. The works [41, 42] discuss the *overlapped* and *latent* trace norm regularizations for tensors. In particular, [42, 45] show that the latent trace norm has certain better tensor reconstruction bounds. The latent trace norm regularization learns the tensor as a *sparse* combination of different tensors. In our work, we empirically motivate the need for learning non-sparse combination of tensors and propose a variant of the latent trace norm that learns a *non-sparse* combination of tensors. We show a novel characterization of the solution space that allows for a compact storage of the tensor, thereby allowing to develop scalable optimization formulations. Concretely, we make the following contributions in this paper.

- We propose a novel trace norm regularizer for low-rank tensor completion problem, which learns a tensor as a non-sparse combination of tensors. In contrast, the more popular latent trace norm regularizer [41, 42, 45] learns a highly sparse combination of tensors. Non-sparse combination helps in capturing information along all the modes.

- We propose a dual framework for analyzing the problem formulation. This provides interesting insights into the solution space of the tensor completion problem, e.g., how the solutions along different modes are related, allowing a compact representation of the tensor.
- Exploiting the characterization of the solution space, we develop a fixed-rank formulation. Our optimization problem is on Riemannian spectrahedron manifolds and we propose computationally efficient trust-region algorithm for our formulation.

Numerical comparisons on real-world datasets for different applications such as video and hyperspectral-image completion, link prediction, and movie recommendation show that the proposed algorithm outperforms state-of-the-art latent trace norm regularized algorithms. The proofs of all the theorems and lemmas and additional experimental details are provided in the longer version of the paper [32]. Our codes are available at `https://pratikjawanpuria.com/`.

## 2 Related work

**Trace norm regularized tensor completion formulations.** The works [27, 42, 37, 34, 9] discuss the *overlapped trace norm* regularization for tensor learning. The overlapped trace norm is motivated as a convex proxy for minimizing the *Tucker* (multilinear) rank of a tensor. The overlapped trace norm is defined as: $R(\mathcal{W}) := \sum_{k=1}^{K} \|\mathcal{W}_k\|_*$, where $\mathcal{W}_k$ is the mode-$k$ matrix unfolding of the tensor $\mathcal{W}$ [25] and $\|\cdot\|_*$ denotes the trace norm regularizer. $\mathcal{W}_k$ is a $n_k \times \Pi_{j \neq k} n_j$ matrix obtained by concatenating mode-$k$ fibers (column vectors) of the form $\mathcal{W}_{(i_1,...,i_{k-1},:,i_{k+1},...,i_K)}$ [25].

*Latent trace norm* is another convex regularizer used for low-rank tensor learning [41, 43, 42, 45, 17]. In this setting, the tensor $\mathcal{W}$ is modeled as *sum* of $K$ (unknown) tensors $\mathcal{W}^{(1)}, \ldots, \mathcal{W}^{(K)}$ such that $\mathcal{W}_k^{(k)}$ are low-rank matrices. The latent trace norm is defined as:

$$R(\mathcal{W}) := \inf_{\sum_{k=1}^{K} \mathcal{W}^{(k)} = \mathcal{W}; \, \mathcal{W}^{(k)} \in \mathbb{R}^{n_1 \times \ldots \times n_K}} \sum_{k=1}^{K} \|\mathcal{W}_k^{(k)}\|_*, \tag{2}$$

A variant of the latent trace norm ($\|\mathcal{W}_k^{(k)}\|_*$ scaled by $1/\sqrt{n_k}$) is analyzed in [45]. Latent trace norm and its scaled variant achieve better recovery bounds than overlapped trace norm [42, 45]. Recently, [17] proposed a scalable latent trace norm based Frank-Wolfe algorithm for tensor completion.

The latent trace norm (2) corresponds to the sparsity inducing $\ell_1$-norm penalization across $\|\mathcal{W}_k^{(k)}\|_*$. Hence, it learns $\mathcal{W}$ as a sparse combination of $\mathcal{W}^{(k)}$. In case of high sparsity, it may result in selecting only one of the tensors $\mathcal{W}^{(k)}$ as $\mathcal{W}$, i.e., $\mathcal{W} = \mathcal{W}^{(k)}$ for some $k$, in which case $\mathcal{W}$ is essentially learned as a low-rank matrix. In several real-world applications, tensor data cannot be mapped to a low-rank matrix structure and they require a higher order structure. Therefore, we propose a regularizer which learns a non-sparse combination of $\mathcal{W}^{(k)}$. Non-sparse norms have led to better generalization performance in other machine learning settings [12, 38, 22].

We show the benefit of learning a non-sparse mixture of tensors as against a sparse mixture on two datasets: Ribeira and Baboon (refer Section 5 for details). Figures 1(a) and 1(b) show the relative sparsity of the optimally learned tensors in the mixture as learned by the $\ell_1$-regularized latent trace norm based model (2) [42, 45, 17] versus the proposed $\ell_2$-regularized model (discussed in Section 3). The relative sparsity for each $\mathcal{W}^{(k)}$ in the mixture is computed as $\|\mathcal{W}^{(k)}\|_F / \sum_k \|\mathcal{W}^{(k)}\|_F$. In both the datasets, our model learns a non-sparse combination of tensors, whereas the latent trace norm

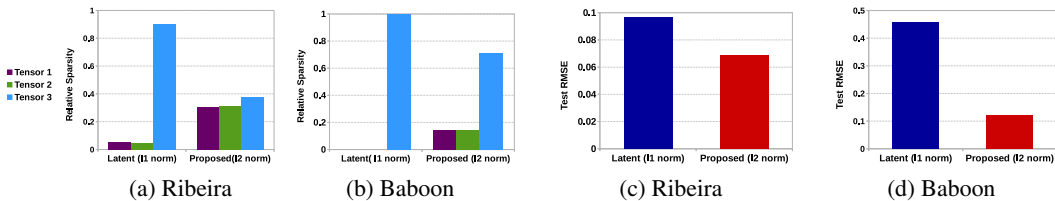

| (a) Ribeira | (b) Baboon | (c) Ribeira | (d) Baboon |

Figure 1: (a) & (b) Relative sparsity of each tensor in the mixture of tensors for Ribeira and Baboon datasets. Our proposed formulation learns a $\ell_2$-norm based non-sparse combination of tensors; (c) & (d) show that the proposed non-sparse combination obtain better generalization performance on both the datasets.

based model learns a highly skewed mixture of tensors. The proposed non-sparse tensor combination also leads to better generalization performance, as can be observed in the Figures 1(c) and 1(d). In the particular case of Baboon dataset, the latent trace norm essentially learns $\mathcal{W}$ as a low-rank matrix ($\mathcal{W} = \mathcal{W}^{(3)}$) and consequently obtains poor generalization.

**Other tensor completion formulations.** Other approaches for low-rank tensor completion include tensor decomposition methods like Tucker and CP [25, 10, 11]. They generalize the notion of singular value decomposition of matrices to tensors. Recently, [26] exploits the Riemannian geometry of fixed multilinear rank to learn factor matrices and the core tensor. They propose a computationally efficient non-linear conjugate gradient method for optimization over manifolds of tensors of fixed multilinear rank. [24] further propose an efficient preconditioner for low-rank tensor learning with the Tucker decomposition. [49] propose a Bayesian probabilistic CP model for performing tensor completion. Tensor completion algorithms based on tensor *tubal-rank* have been recently proposed in [48, 28].

## 3 Non-sparse latent trace norm and duality

We propose the following formulation for learning the low-rank tensor $\mathcal{W}$

$$\min_{\mathcal{W}^{(k)} \in \mathbb{R}^{n_1 \times \dots \times n_K}} \left\| \sum_k \mathcal{W}^{(k)}_\Omega - \mathcal{Y}_\Omega \right\|^2_F + \sum_k \frac{1}{\lambda_k} \|\mathcal{W}^{(k)}_k\|^2_*, \tag{3}$$

where $\mathcal{W} = \sum_k \mathcal{W}^{(k)}$ is the learned tensor. It should be noted that the proposed regularizer in (3) employs the $\ell_2$-norm over $\|\mathcal{W}^{(k)}_k\|_*$. In contrast, the latent trace norm regularizer (2) has the $\ell_1$-norm over $\|\mathcal{W}^{(k)}_k\|_*$.

While the existing tensor completion approaches [24, 17, 26, 27, 42, 37] mostly discuss a primal formulation similar to (1), we propose a novel dual framework for our analysis. The use of dual framework for learning low-rank matrices [46, 20], multi-task problems [33, 21, 19], *etc.*, often leads to novel insights into the solution space of the primal problem.

We begin by discussing how to obtain the dual formulation of (3). Later, we explain how the insights from the dual framework motivate us to propose a novel fixed-rank formulation. As a first step, we exploit the following *variational characterization* of the trace norm studied in [3, Theorem 4.1]. Given $\mathbf{X} \in \mathbb{R}^{d \times T}$, the following result holds:

$$\|\mathbf{X}\|^2_* = \min_{\Theta \in \mathcal{P}^d, \text{range}(\mathbf{X}) \subseteq \text{range}(\Theta)} \langle \Theta^\dagger, \mathbf{X}\mathbf{X}^\top \rangle, \tag{4}$$

where $\mathcal{P}^d$ denotes the set of $d \times d$ positive semi-definite matrices with *unit trace*, $\Theta^\dagger$ denotes the pseudo-inverse of $\Theta$, $\text{range}(\Theta) = \{\Theta z : z \in \mathbb{R}^d\}$, and $\langle \cdot, \cdot \rangle$ is the inner product. The expression for optimal $\Theta^*$ is $\Theta^* = \sqrt{\mathbf{X}\mathbf{X}^\top}/\text{trace}(\sqrt{\mathbf{X}\mathbf{X}^\top})$ [3], and hence the ranks of $\Theta$ and $\mathbf{X}$ are equal at optimality. Thus, (4) implicitly transfers the low-rank constraint on $\mathbf{X}$ (due to trace norm) to an auxiliary variable $\Theta \in \mathcal{P}^d$. It is well known that positive semi-definite matrix $\Theta$ with unit trace constraint implies the $\ell_1$-norm constraint on the eigenvalues of $\Theta$, leading to low-rankedness of $\Theta$. The result (4) has also been recently employed to obtain new factorization insights for structured low-rank matrices [20].

Using the result (4) in (3) leads to $K$ auxiliary matrices, one $\Theta_k \in \mathcal{P}^{n_k}$ corresponding to every $\mathcal{W}^{(k)}_k$ (mode-$k$ matrix unfolding of the tensor $\mathcal{W}^{(k)}$). It should also be noted that $\Theta_k \in \mathcal{P}^{n_k}$ are low-rank matrices. We now present the following theorem that states an equivalent minimax formulation of (3).

**Theorem 1** *An equivalent minimax formulation of the problem (3) is*

$$\min_{\Theta_1 \in \mathcal{P}^{n_1}, \dots, \Theta_K \in \mathcal{P}^{n_K}} \max_{\mathcal{Z} \in \mathcal{C}} \langle \mathcal{Z}, \mathcal{Y}_\Omega \rangle - \frac{1}{4}\|\mathcal{Z}\|^2_F - \sum_k \frac{\lambda_k}{2} \langle \Theta_k, \mathcal{Z}_k \mathcal{Z}^\top_k \rangle, \tag{5}$$

*where $\mathcal{Z}$ is the dual tensor variable corresponding to the primal problem (3) and $\mathcal{Z}_k$ is the mode-$k$ unfolding of $\mathcal{Z}$. The set $\mathcal{C} := \{\mathcal{Z} \in \mathbb{R}^{n_1 \times \dots \times n_K} : \mathcal{Z}_{(i_1, \dots, i_K)} = 0, (i_1, \dots, i_K) \notin \Omega\}$ constrains $\mathcal{Z}$ to be a sparse tensor with $|\Omega|$ non-zero entries. Let $\{\Theta^*_1, \dots, \Theta^*_K, \mathcal{Z}^*\}$ be the optimal solution of (5). The optimal solution of (3) is given by $\mathcal{W}^* = \sum_k \mathcal{W}^{(k)*}$, where $\mathcal{W}^{(k)*} = \lambda_k(\mathcal{Z}^* \times_k \Theta^*_k) \forall k$ and $\times_k$ denotes the tensor-matrix multiplication along mode $k$.*

---
**Algorithm 1** Proposed Riemannian trust-region algorithm for (7).
---
**Input:** $\mathcal{Y}_\Omega$, rank $(r_1, \ldots, r_K)$, regularization parameter $\lambda$, and tolerance $\epsilon$.
**Initialize :** $u \in \mathcal{M}$.
**repeat**
    **1:** Compute the gradient $\nabla_u \ell$ for (7) as given in Lemma 1.
    **2:** Compute the search direction which minimizes the trust-region subproblem.
       It makes use of $\nabla_u \ell$ and its directional derivative presented in Lemma 1 for (7).
    **3:** Update $x$ with the retraction step to maintain strict feasibility on $\mathcal{M}$. Specifically for the
       spectrahedron manifold, $\mathbf{U}_k \leftarrow (\mathbf{U}_k + \mathbf{V}_k) / \|\mathbf{U}_k + \mathbf{V}_k\|_F$, where $\mathbf{V}_k$ is the search direction.
**until** $\|\nabla_u \ell\|_F < \epsilon$.
**Output:** $u^*$
---

**Remark 1:** Theorem 1 shows that the optimal solutions $\mathcal{W}^{(k)*}$ for all $k$ in (3) are completely charac-terized by a *single* sparse tensor $\mathcal{Z}^*$ and $K$ low-rank positive semi-definite matrices $\{\Theta_1^*, \ldots, \Theta_K^*\}$. It should be noted that such a novel relationship of $\mathcal{W}^{(k)*}$ (for all $k$) with each other is not evident from the primal formulation (3).

We next present the following result related to the form of the optimal solution of (3).

**Corollary 1** *(Representer theorem) The optimal solution of the primal problem (3) admits a repre-sentation of the form:* $\mathcal{W}^{(k)*} = \lambda_k (\mathcal{Z} \times_k \Theta_k) \, \forall k$, *where* $\mathcal{Z} \in \mathcal{C}$ *and* $\Theta_k \in \mathcal{P}^{n_k}$.

As discussed earlier in the section, the optimal $\Theta_k^* \in \mathcal{P}^{n_k}$ is a low-rank positive semi-definite matrix for all $k$. In spite of the low-rankness of the optimal solution, an algorithm for (5) need not produce intermediate iterates that are low rank. From the perspective of large-scale applications, this observation as well as other computational efficiency concerns discussed below motivate to exploit a fixed-rank parameterization of $\Theta_k$ for all $k$.

**Fixed-rank parameterization.** We propose to explicitly constrain the rank of $\Theta_k$ to $r_k$ as follows:

$$\Theta_k = \mathbf{U}_k \mathbf{U}_k^\top, \tag{6}$$

where $\mathbf{U}_k \in \mathcal{S}_{r_k}^{n_k}$ and $\mathcal{S}_r^n := \{\mathbf{U} \in \mathbb{R}^{n \times r} : \|\mathbf{U}\|_F = 1\}$. In large-scale tensor completion problems, it is common to set $r_k \ll n_k$, where the fixed-rank parameterization (6) of $\Theta_k$ has a two-fold advantage. First, the search space dimension drastically reduces from $n_k((n_k + 1)/2 - 1)$, which is *quadratic* in tensor dimensions, to $n_k r_k - 1 - r_k(r_k - 1)/2$, which is *linear* in tensor dimensions [23]. Second, enforcing the constraint $\mathbf{U}_k \in \mathcal{S}_{r_k}^{n_k}$ costs $O(n_k r_k)$, which is *linear* in tensor dimensions and is computationally much cheaper than enforcing $\Theta_k \in \mathcal{P}^{n_k}$ that costs $O(n_k^3)$.

Employing the proposed fixed-rank parameterization (6), we propose a scalable tensor completion dual formulation.

**Fixed-rank dual formulation.** The first formulation is obtained by employing the parameterization (6) directly in (5). We subsequently solve the resulting problem as a minimization problem as follows:

$$\min_{u \in \mathcal{S}_{r_1}^{n_1} \times \ldots \times \mathcal{S}_{r_K}^{n_K}} g(u), \tag{7}$$

where $u = (\mathbf{U}_1, \ldots, \mathbf{U}_K)$ and $g : \mathcal{S}_{r_1}^{n_1} \times \ldots \times \mathcal{S}_{r_K}^{n_K} \to \mathbb{R}$ is the function

$$g(u) := \max_{\mathcal{Z} \in \mathcal{C}} \langle \mathcal{Z}, \mathcal{Y}_\Omega \rangle - \frac{1}{4} \|\mathcal{Z}\|_F^2 - \sum_k \frac{\lambda_k}{2} \left\| \mathbf{U}_k^\top \mathcal{Z}_k \right\|_F^2. \tag{8}$$

It should be noted that though (7) is a non-convex problem in $u$, the optimization problem in (8) is *strongly* convex in $\mathcal{Z}$ for a given $u$ and has *unique* solution.

## 4 Optimization algorithm

The optimization problem (7) is of the form

$$\min_{x \in \mathcal{M}} \ell(x), \tag{9}$$

where $\ell : \mathcal{M} \to \mathbb{R}$ is a smooth loss and $\mathcal{M} := \mathcal{S}_{r_1}^{n_1} \times \ldots \times \mathcal{S}_{r_K}^{n_K} \times \mathcal{C}$ is the constraint set.

In order to propose numerically efficient algorithms for optimization over $\mathcal{M}$, we exploit the particular structure of the set $\mathcal{S}_r^n$, which is known as the *spectrahedron* manifold [23]. The spectrahedron manifold has the structure of a compact Riemannian quotient manifold [23]. Consequently, optimization on the spectrahedron manifold is handled in the Riemannian optimization framework. This allows to exploit the rotational invariance of the constraint $\|\mathbf{U}\|_F = 1$ naturally. The Riemannian manifold optimization framework embeds the constraint $\|\mathbf{U}\|_F = 1$ into the search space, thereby translating the constrained optimization problem into *unconstrained* optimization problem over the spectrahedron manifold. The Riemannian framework generalizes a number of classical first- and second-order (e.g., the conjugate gradient and trust-region algorithms) Euclidean algorithms to manifolds and provides concrete convergence guarantees [13, 1, 36, 47, 35]. The work [1], in particular, shows a systematic way of implementing trust-region (TR) algorithms on quotient manifolds. A full list of optimization-related ingredients and their matrix characterizations for the spectrahedron manifold $\mathcal{S}_r^n$ is in the supplementary material. Overall, the constraint $\mathcal{M}$ is endowed a Riemannian structure.

We implement the Riemannian TR (second-order) algorithm for (9). To this end, we require the notions of the *Riemannian gradient* (the first-order derivative of the objective function on the manifold), the *Riemannian Hessian* along a search direction (the *covariant* derivative of the Riemannian gradient along a tangential direction on the manifold), and the *retraction* operator which ensures that we always stay on the manifold (i.e., maintain strict feasibility). The Riemannian gradient and Hessian notions require computations of the standard (Euclidean) gradient $\nabla_x \ell(x)$ and the directional derivative of this gradient along a given search direction $v$ denoted by $\mathrm{D}\nabla_x \ell(x)[v]$. The expressions of both for (7) are given in Lemma 1.

**Lemma 1** *Let $\hat{\mathcal{Z}}$ be the optimal solution of the convex problem (8) at $u \in \mathcal{S}_{r_1}^{n_1} \times \ldots \times \mathcal{S}_{r_K}^{n_K}$. Let $\nabla_u g$ denote the gradient of $g(u)$ at $u$, $\mathrm{D}\nabla_u g[v]$ denote the directional derivative of the gradient $\nabla_u g$ along $v \in \mathbb{R}^{n_1 \times r_1} \times \ldots \times \mathbb{R}^{n_K \times r_K}$, and $\dot{\mathcal{Z}}_k$ be the directional derivative of $\mathcal{Z}_k$ along $v$ at $\hat{\mathcal{Z}}_k$. Then, $\nabla_u g = (-\lambda_1 \hat{\mathcal{Z}}_1 \hat{\mathcal{Z}}_1^\top \mathbf{U}_1, \ldots, -\lambda_K \hat{\mathcal{Z}}_K \hat{\mathcal{Z}}_K^\top \mathbf{U}_K)$, and $\mathrm{D}\nabla_u g[v] = (-\lambda_1 \mathbf{A}_1, \ldots, -\lambda_K \mathbf{A}_K)$, where $\mathbf{A}_k = \hat{\mathcal{Z}}_k \hat{\mathcal{Z}}_k^\top \mathbf{V}_k + \mathrm{symm}(\dot{\mathcal{Z}}_k \hat{\mathcal{Z}}_k^\top) \mathbf{U}_k$ and $\mathrm{symm}(\boldsymbol{\Delta}) = (\boldsymbol{\Delta} + \boldsymbol{\Delta}^\top)/2$.*

A key requirement in Lemma 1 is to efficiently solve (8) for a given $u = (\mathbf{U}_1, \ldots, \mathbf{U}_K)$. It should be noted that (8) has a closed-form sparse solution, which is equivalent to solving the *linear* system

$$\hat{\mathcal{Z}}_\Omega + \sum_k \lambda_k (\hat{\mathcal{Z}}_\Omega \times_k \mathbf{U}_k \mathbf{U}_k^\top)_\Omega = \mathcal{Y}_\Omega. \tag{10}$$

Solving the linear system (10) in a *single* step is computationally expensive (it involves the use of Kronecker products, vectorization of a sparse tensor, and a matrix inversion). Instead, we use an *iterative* solver that exploits the sparsity in the variable $\mathcal{Z}$ and the factorization form $\mathbf{U}_k \mathbf{U}_k^\top$ efficiently. Similarly, given $\hat{\mathcal{Z}}$ and $v$, $\dot{\mathcal{Z}}$ can be computed by solving

$$\dot{\mathcal{Z}}_\Omega + \sum_k \lambda_k (\dot{\mathcal{Z}}_\Omega \times_k \mathbf{U}_k \mathbf{U}_k^\top)_\Omega = -\sum_k \lambda_k (\hat{\mathcal{Z}}_\Omega \times_k (\mathbf{V}_k \mathbf{U}_k^\top + \mathbf{U}_k \mathbf{V}_k^\top))_\Omega. \tag{11}$$

The Riemannian TR algorithm solves a Riemannian trust-region sub-problem in every iteration [1, Chapter 7]. The TR sub-problem is a *second-order* approximation of the objective function in a neighborhood, solution to which does not require inverting the full Hessian of the objective function. It makes use of the gradient $\nabla_x \ell$ and its directional derivative along a search direction. The TR sub-problem is approximately solved with an iterative solver, e.g., the truncated conjugate gradient algorithm. The TR sub-problem outputs a potential update candidate for $x$, which is then accepted or rejected based on the amount of decrease in the function $\ell$. Algorithm 1 summarizes the key steps of the TR algorithm for solving (9).

**Computational complexity:** the per-iteration computational complexity of Algorithm 1 scales linearly with the number of known entries $\mathcal{Y}_\Omega$, denoted by $|\Omega|$. In particular, the per-iteration computational cost depends on the following ingredients.

- $\mathbf{U}_k^\top \mathcal{Z}_k$: it involves computation of $n_k \times r_k$ matrix $\mathbf{U}_k$ with mode-$k$ unfolding of a sparse $\mathcal{Z}$ with $|\Omega|$ non-zero entries. This costs $O(|\Omega| r_k)$. It should be noted that although the dimension of $\mathcal{Z}_k$ is $n_k \times \prod_{i=1, i \neq k}^{K} n_i$, only a maximum of $|\Omega|$ columns have non-zero entries. We exploit this property of $\mathcal{Z}_k$ and have a compact memory storage of $\mathbf{U}_k^\top \mathcal{Z}_k$.

- Computing the solution $\hat{\mathcal{Z}}$ of the linear system (10): an iterative solver for (10) requires computing the left hand side of (10) for a given candidate $\mathcal{Z}$. This costs $O(|\Omega| \sum_k r_k)$.

Table 1: Summary of the baseline low-rank tensor completion algorithms.

| Trace norm regularized algorithms | | Other algorithms | |
|---|---|---|---|
| FFW | Scaled latent trace norm + Frank Wolfe optimization + basis size reduction | Topt | Fixed multilinear rank + conjugate gradients (CG) |
| Hard | Scaled overlapped trace norm + proximal gradient | BayesCP | Bayesian CP algorithm with rank tuning |
| | | geomCG | Riemannian CG + fixed multilinear rank |
| HaLRTC | Scaled overlapped trace norm + ADMM | Rprecon | Riemannian CG with preconditioning + fixed multilinear rank |
| Latent | Latent trace norm + ADMM | T-svd | Tensor tubal-rank + ADMM |

- Computation of $g(u)$: it relies on the solution of (10) and then explicitly computing the objective function in (8). This costs $O(|\Omega| \sum_k r_k + K|\Omega|)$.

- $\nabla_u g(u)$: it requires the computation of terms like $\hat{\boldsymbol{\mathcal{Z}}}_k(\hat{\boldsymbol{\mathcal{Z}}}_k^\top \mathbf{U}_k)$, which cost $O(|\Omega| \sum_k r_k)$.

- Computing the solution $\dot{\boldsymbol{\mathcal{Z}}}_k$ of the linear system (11): similar to (10), (11) is solved with an iterative solver. The computational cost of solving (11) requires computing terms like $\mathbf{U}_k^\top \boldsymbol{\mathcal{Z}}_k$ and $\mathbf{U}_k^\top \dot{\boldsymbol{\mathcal{Z}}}_k$, which costs $O(|\Omega| r_k)$. It should be noted that both $\dot{\boldsymbol{\mathcal{Z}}}$ and $\hat{\boldsymbol{\mathcal{Z}}}$ share the same sparsity pattern.

- $D\nabla_u g(u)[v]$: it costs $O(|\Omega| \sum_k r_k)$.

- Retraction on $\mathcal{S}_{r_k}^{n_k}$: it projects a matrix of size $n_k \times r_k$ on to the set $\mathcal{S}_{r_k}^{n_k}$, which costs $O(n_k r_k)$.

- $\mathcal{S}_{r_k}^{n_k}$ manifold-related ingredients cost $O(n_k r_k^2 + r_k^3)$.

Overall, the per-iteration computational complexities of our algorithm is $O(m(|\Omega| \sum_k r_k + \sum_k n_k r_k^2 + \sum_k r_k^3))$, where $m$ is the number of iterations needed to solve (10) and (11) approximately. The memory cost for our algorithm is $O(|\Omega| + \sum_k n_k r_k)$. We observe that both computational and memory cost scales *linearly* with the number of observed entries ($|\Omega|$), which makes our algorithms scalable to large datasets.

**Convergence.** The Riemannian TR algorithms come with rigorous convergence guarantees. [1] discuss the rate of convergence analysis of manifold algorithms, which directly apply in our case. For trust regions, the global convergence to a first-order critical point is discussed in [1, Section 7.4.1] and the local convergence to local minima is discussed in [1, Section 7.4.2]. From an implementation perspective, we follow the existing approaches [26, 24, 17] and bound the number of TR iterations.

**Numerical implementation.** Our algorithm is implemented using the Manopt toolbox [7] in Matlab, which has off-the-shelf generic TR implementation.

## 5    Experiments

We evaluate the generalization performance and efficiency of our proposed TR algorithm against state-of-the-art algorithms in several tensor completion applications.

**Trace norm regularized algorithms**. Scaled latent trace norm regularized algorithms such as **FFW** [17] and **Latent** [42], and overlapped trace norm based algorithms such as **HaLRTC** [27] and **Hard** [37] are the closest to our approach. FFW is a recently proposed state-of-the-art large scale tensor completion algorithm. Table 1 summarizes the trace norm regularized baseline algorithms.

We denote our algorithm as **TR-MM** (**T**rust-**R**egion algorithm for **M**ini**M**ax tensor completion formulation). We set $\lambda_k = \lambda n_k \ \forall k$ in (7). Hence, we tune only one hyper-parameter $\lambda$, from the set $\{10^{-3}, 10^{-2}, \ldots, 10^3\}$, via five-fold cross-validation of the training data.

**Video and image completion**

We work with the following datasets for predicting missing values in multi-media data: a) **Ribeira** is a hyperspectral image [16] of size $1017 \times 1340 \times 33$, where each slice represents the image measured at a particular wavelength. We re-size it to $203 \times 268 \times 33$ [37, 26, 24]; b) **Tomato** is a video sequence dataset [27, 8] of size $242 \times 320 \times 167$; and c) **Baboon** is an RGB image [49], modeled as

Table 2: Generalization performance across several applications: hyperspectral-image/video/image completion, movie recommendation, and link prediction. Our algorithm, TR-MM, performs significantly better than other trace norm based algorithms and obtain the best overall performance. The symbol '−' denotes the dataset is too large for the algorithm to generate result.

|  | TR-MM | FFW | Rprecon | geomCG | Hard | Topt | HaLRTC | Latent | T-svd | BayesCP |
|---|---|---|---|---|---|---|---|---|---|---|
| RMSE reported |  |  |  |  |  |  |  |  |  |  |
| Ribeira | 0.067 | 0.088 | 0.083 | 0.156 | 0.114 | 0.127 | 0.095 | 0.087 | **0.064** | 0.154 |
| Tomato | **0.041** | 0.045 | 0.052 | 0.052 | 0.060 | 0.102 | 0.202 | 0.046 | 0.042 | 0.103 |
| Baboon | **0.121** | 0.133 | 0.128 | 0.128 | 0.126 | 0.130 | 0.247 | 0.459 | 0.146 | 0.159 |
| ML10M | 0.840 | 0.895 | **0.831** | 0.844 | – | – | – | – | – | – |
| AUC reported |  |  |  |  |  |  |  |  |  |  |
| YouTube (subset) | **0.957** | 0.954 | 0.941 | 0.941 | 0.954 | 0.941 | 0.783 | 0.945 | 0.941 | 0.950 |
| YouTube (full) | **0.932** | 0.929 | 0.926 | 0.926 | – | – | – | – | – | – |
| FB15k-237 | **0.823** | 0.764 | 0.821 | 0.785 | – | – | – | – | – | – |

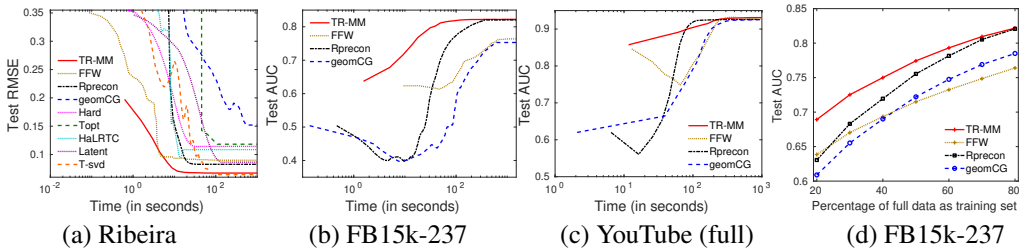

| (a) Ribeira | (b) FB15k-237 | (c) YouTube (full) | (d) FB15k-237 |
|---|---|---|---|

Figure 2: (a) Evolution of test RMSE on Ribeira; (b) & (c) Evolution of test AUC on FB15k-237 and YouTube (full), respectively. Our algorithm, TR-MM, obtains the best generalization performance in all the three datasets. In addition, TR-MM converges to a good solution is fairly quick time; (d) Variation of test AUC as the amount of training data changes on FB15k-237. TR-MM performs significantly better than baselines when the amount of training data is less.

a $256 \times 256 \times 3$ tensor. Following [24], we train on a random sample of $10\%$ of the entries and test on another $10\%$ of the entries for all the three datasets. Each experiment is repeated ten times.

**Results**. Table 2 reports the root mean squared error (RMSE) on the test set, averaged over ten splits. Our algorithm, TR-MM, obtains the best results, outperforming other trace norm based algorithms on all the three datasets. Figure 2(a) shows the trade-off between the test RMSE and the training time of all the algorithms on Ribeira. It can be observed that TR-MM converges to the lowest RMSE at a significantly faster rate compared to the other baselines. It is evident from the results that learning a mixture of non-sparse tensors, as learned by the proposed algorithm, helps in achieving better generalization performance compared to the algorithms that learn a sparse mixture of tensors.

### Link prediction

The aim in link prediction setting is to predict missing or new links in knowledge graphs, social networks, *etc*. We consider FB15k-237 and YouTube datasets, discussed below.

**FB15k-237:** this is a subset of FB15k dataset [6, 44], containing facts of the form subject-predicate-object (RDF) triples from Freebase knowledge graph. FB15k-237 contains $14\,541$ entities and $237$ relationships. The task is to predict the relationships (from a given set of relations) between a pair of entities in the knowledge graph. It has $310\,116$ observed relationships (links) between pairs of entities, which are the *positive* samples. In addition, $516\,606$ *negative* samples are generated following the procedure described in [6]. We model this task as a $14\,541 \times 14\,541 \times 237$ tensor completion problem. $\mathcal{Y}_{(a,b,c)} = 1$ implies that relationship$_b$ exists between entity$_a$ and entity$_c$, and $\mathcal{Y}_{(a,b,c)} = 0$ implies otherwise. We keep $80\%$ of the observed entries for training and the remaining $20\%$ for testing.

**YouTube:** this is a link prediction dataset [40] having 5 types of interactions between $15\,088$ users. The task is to predict the interaction (from a given set of interactions) between a pair of users. We model it as a $15\,088 \times 15\,088 \times 5$ tensor completion problem. All the entries are known in this case. We randomly sample $0.8\%$ of the data for training [17] and another $0.8\%$ for testing.

Table 3: Rank sets at which the proposed TR-MM algorithm and the Tucker decomposition based tensor completion algorithms (Rprecon, geomGC, Topt) achieve best results across datasets. It should be noted that the notion of rank in trace norm regularized approaches (such as TR-MM) differs from the Tucker rank.

|  | TR-MM rank | Tucker rank |
| --- | --- | --- |
| Ribeira | $(5, 5, 5)$ | $(15, 15, 6)$ |
| Tomato | $(10, 10, 10)$ | $(15, 15, 15)$ |
| Baboon | $(4, 4, 3)$ | $(4, 4, 3)$ |
| ML10M | $(20, 10, 1)$ | $(4, 4, 4)$ |
| YouTube (subset) | $(3, 3, 1)$ | $(5, 5, 5)$ |
| YouTube (full) | $(3, 3, 1)$ | $(5, 5, 5)$ |
| FB15k-237 | $(20, 20, 1)$ | $(5, 5, 5)$ |

It should be noted that Hard, HaLRTC, and Latent do not scale to the full FB15k-237 and YouTube datasets as they need to store full tensor in memory. Hence, we follow [17] to create a subset of the YouTube dataset of size $1509 \times 1509 \times 5$ in which 1509 users with most number of links are chosen. We randomly sample $5\%$ of the data for training and another $5\%$ for testing.

Each experiment is repeated on ten random train-test splits. Following [29, 17], the generalization performance for link prediction task is measured by computing the area under the ROC curve on the test set (test AUC) for each algorithm.

**Results**. Table 2 reports the average test AUC on YouTube (subset), Youtube (full) and FB15k-237 datasets. The TR-MM algorithm achieves the best performance in all the link prediction tasks. This shows that the non-sparse mixture of tensors learned by TR-MM helps in achieving better performance. Figures 2(b) & 2(c) plots the trade-off between the test AUC and the training time for FB15k-237 and YouTube, respectively. We observe that TR-MM is the fastest to converge to a good AUC and take only a few iterations.

We also conduct experiments to evaluate the performance of different algorithms in challenging scenarios when the amount of training data available is less. On the FB15k-237 dataset, we vary the size of training data from $20\%$ to $80\%$ of the observed entries, and the remaining $20\%$ of the observed entries is kept as the test set. Figure 2(d) plots the results of this experiment. We can observe that TR-MM does significantly better than the baselines in data scarce regimes.

### Movie recommendation

We evaluate the algorithms on the MovieLens10M (ML10M) dataset [18]. This is a movie recommendation task — predict the ratings given to movies by various users. MovieLens10M contains $10\,000\,054$ ratings of $10\,681$ movies given by $71\,567$ users. Following [24], we split the time into 7-days wide bins, forming a tensor of size $71\,567 \times 10\,681 \times 731$. For our experiments, we generate ten random train-test splits, where $80\%$ of the observed entries is kept for training and the rest $20\%$ for testing.

**Results**. Table 2 reports the average test RMSE on this task. It can be observed that our algorithm, TR-MM, outperforms state-of-the-art scaled latent trace norm based algorithm FFW.

### Results compared to other baseline algorithms

In addition to the trace norm based algorithms, we also compare against algorithms that model tensor via Tucker decomposition with fixed multilinear ranks: **Rprecon** [24], **geomCG** [26], and **Topt** [15]. Large scale state-of-the-art algorithms in this multilinear framework include Rprecon and geomCG. We also compare against tensor tubal-rank based algorithm **T-svd** [48] and CP decomposition based algorithm **BayesCP** [49]. Table 1 summarizes these baselines.

As can be observed from Table 2, TR-MM obtains better overall generalization performance than the above discussed baselines. In the movie recommendation problem, Rprecon achieves better results than TR-MM. It should be noted that Topt, T-svd, and BayesCP are not scalable to large scale datasets.

### Rank of solutions of TR-MM algorithm

Table 3 shows the rank sets at which the proposed TR-MM and Tucker decomposition based tensor completion algorithms (Rprecon, geomGC, Topt) achieve best results across datasets. The latent

Table 4: Results on outlier robustness experiments. Our algorithm, TR-MM, is more robust to outliers than the competing baselines. The symbol '$-$' denotes the dataset is too large for the algorithm to generate result.

| | $x$ | TR-MM | FFW | Rprecon | geomCG | Hard | Topt | HaLRTC | Latent | T-svd | BayesCP |
|---|---|---|---|---|---|---|---|---|---|---|---|
| Ribeira (RMSE) | 0.05 | **0.081** | 0.095 | 0.157 | 0.258 | 0.142 | 0.169 | 0.121 | 0.103 | 0.146 | 0.201 |
| | 0.10 | **0.111** | 0.112 | 0.172 | 0.373 | 0.158 | 0.188 | 0.135 | 0.120 | 0.182 | 0.204 |
| FB15k-237 (AUC) | 0.05 | **0.803** | 0.734 | 0.794 | 0.764 | $-$ | $-$ | $-$ | $-$ | $-$ | $-$ |
| | 0.10 | **0.772** | 0.711 | 0.765 | 0.739 | $-$ | $-$ | $-$ | $-$ | $-$ | $-$ |

trace norm based algorithms (TR-MM, FFW, Latent) model the tensor completion by approximating the input tensor as a combination of tensors. Each tensor in this combination is constrained to be low-ranked along a given mode. In contrast, Tucker-decomposition based algorithms model the tensor completion problem as a factorization problem with the given Tucker rank (also known as the multilinear rank). Due to this fundamental difference in modeling, the concept of rank in TR-MM algorithm is different from the multilinear rank of Tucker decomposition based algorithms.

**Results on outlier robustness**

We also evaluate TR-MM and the baselines considered in Table 1 for outlier robustness on hyper-spectal image completion and link prediction problems. In the Ribeira dataset, we add the standard Gaussian noise ($N(0, 1)$) to randomly selected $x$ fraction of the entries in the training set. The minimum and the maximum value of entries in the (original) Ribeira are 0.01 and 2.09, respectively. In FB15k-237 dataset, we flip randomly selected $x$ fraction of the entries in the training set, *i.e.*, the link is removed if present and vice-versa. We experiment with $x = 0.05$ and $x = 0.10$.

The results are reported in Table 4. We observe that our algorithm, TR-MM, obtains the best generalization performance and, hence, is the most robust to outliers. We also observe that trace norm regularized algorithms are relatively more robust to outliers than Tucker-decomposition, CP-decomposition, and tensor tubal-rank based algorithms.

## 6    Discussion and conclusion

In this paper, we introduce a novel regularizer for low-rank tensor completion problem which learns the tensor as a non-sparse combination of $K$ tensors, where $K$ is the number of modes. Existing works [41, 42, 45, 17] learn a sparse combination of tensors, essentially learning the tensor as a low-rank matrix and losing higher order information in the available data. Hence, we recommend learning a non-sparse combination of tensors in trace norm regularized setting, especially since $K$ is typically a small integer in most real-world applications. In our experiments, we observe better generalization performance with the proposed regularization. Theoretically, we provide the following result on the reconstruction error in the context of recovering an unknown tensor $\mathcal{W}^*$ from noisy observation (a similar result on the *latent trace norm* is presented in [42]).

**Lemma 2** *Let $\mathcal{W}^*$ be the true tensor to be recovered from observed $\mathcal{Y}$, which is obtained as $\mathcal{Y} = \mathcal{W}^* + \mathcal{E}$, where $\mathcal{E} \in \mathbb{R}^{n_1 \times \cdots \times n_K}$ is the noise tensor. Assume that the regularization constant $\lambda$ satisfies $\lambda \leq 1/(\sum_{k=1}^{K} \|\mathcal{E}_k\|_\infty^2)^{1/2}$ then the estimator*

$$\hat{\mathcal{W}} = \underset{\mathcal{W}}{argmin}(\frac{1}{2}\|\mathcal{Y} - \mathcal{W}\|_F^2 + \frac{1}{\lambda}\sum_k \|\mathcal{W}_k^{(k)}\|_*^2),$$

*satisfies the inequality $\|\hat{\mathcal{W}} - \mathcal{W}^*\|_F \leq \frac{2}{\lambda}\sqrt{\min_k n_k}$. When noise approaches zero, i.e., $\mathcal{E} \to 0$, the right hand side also approaches zero.*

We present a dual framework to analyze the proposed tensor completion formulation. This leads to a novel fixed-rank formulation, for which we exploit the Riemannian framework to develop scalable trust region algorithm. In experiments, our algorithm TR-MM obtains better generalization performance and is more robust to outliers than state-of-the-art low-rank tensor completion algorithms.

In future, optimization algorithms for the proposed formulation can be developed for online or distributed frameworks. Recent works [4, 5, 30, 31] have explored optimization over Riemannian manifolds in such learning settings.

## Acknowledgement

Most of this work was done when MN (as an intern), PJ, and BM were at Amazon.

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
