[Reviews · NeurIPS 2018]

Reviewer 1



The article focus on tensor completion by trace-norm regularization. In this approach, tensor with missing data are completed with a decomposition as a sum of K low-rank tensors. The rank of each of these tensor components is not fixed but constrained by a trace-norm regularization on the specific matricized form of each tensor (so tensor component K will be constrained to have a low-rank k-mode matricization). The main contribution is to propose another regularization than latent-trace norm which tends to eliminate components, thus enabling a non-sparse decomposition. The proposed trace norm regularizer can be rewritten through variational characterization as the solution of a minimization problem on the space of positive semi-definite matrices with unit trace (defined as the spectrahedron manifold in Journée at al./2008) and two novel formulations whose main interests are to depends on only one dual tensor (compared to K components) and K matrices are presented. As the matrices belongs to the spectrahedron manifold, they can be parametrized through a product of a fixed-rank matrices (whose frobenius norm is 1) with any element of the orthogonal group. As a consequence, the fixed-parameters problems should be solved through optimization on a quotient manifold. Here a Trust-Region (second order) algorithm is chosen which necessitates the computation of the Riemannian gradient, the Riemannian hessian and a retraction operator. All elements are provided and detailed in the supplementary material. Last, several experiments (color image completion, link prediction, movie recommendation) are presented that shows the general efficiency of the method. There are some missing details like the chosen ranks of the parametrization of the spectrahedron matrices which could influence the effectiveness of the method. However as codes are provided, the reproductivity of some of the experiments is possible. My main complaint with this work is: as this works focus in tensor completion, it eschews to bring into the light its relationship with matrix completion methods. In particular, there are a lot of elements in common with the recent (ICML’18) work [P. Jawanpuria and B. Mishra. A Unified framework for structured low-rank matrix learning. International Conference on Machine Learning, 2018.] (cited in the text as [17]). In particular, the dual trick, the main Riemannian components necessary for the Trust Region implementation (projections, retraction) are the same. Another remark is that the use of trace-norm regularization enables to not specify the ranks of each components of the decomposition, whereas the chosen parametrization of the elements of the spectrahedron manifold makes the choice of ranks of the matrix again necessary. A summary is that the addressed problem is interesting, the algorithms proposed make a good use of the Riemannian framework and the experiments are good. The main negative point is that the contributions of this work may rely a lot to similar work on matricial decomposition

Reviewer 2



The authors introduce an improved optimization model to the trace norm regularized tensor completion problem. That model combines some elements of earlier proposed approaches, for example the regularization is a reformulation of the latent trace norm based approach. Via applying square trace norm the authors try to avoid specific sparse optimum solution. This approach resembles to to a maximum entropy type optimization model realized by a Gini concentration based regularization. Soundness: technically correct advanced optimization algorithm Clarity: clearly written, ease to follow Originality: the paper is built upon a synthesis of several known techniques Significance: The manifold based optimization framework sounds very useful Details: The proposed new formulation of the problem is mainly incremental. The real value of the paper is the optimization algorithm which is a combination of different sophisticated techniques. It transforms the original problem into one which is built onto a tensor manifold constraint, where the manifold turns to be a spectrahedron, a Riemannian quotient manifold of positive semidefinite matrices. The reason why this transformation into a manifold represented problem is applied has not been explained in the paper. Could this sophisticated algorithm be reduced into a simpler one which might be even more efficient? The minimax problem in Theorem 1 could be interpreted as a Fenchel duality based reformulation of the original case. Exploiting the theory of that type of duality might lead to a simpler reasoning than that which is included in the paper. I have read the author response. I think the paper contains an advanced exploitation of the known techniques, but at limited level of novelty.

Reviewer 3



[Summary] In this paper, a new tensor completion method is proposed using 'l2' latent trace norm regularization. Authors claimed that the standard(l1) latent trace norm regularization has an issue of ignoring the low-rank structure of several modes, and focusing only on the low-rank structure of a few specific modes. Thus, Authors proposed the l2 latent trace norm regularization. Its dual formulation provides some efficient low-dimensional optimization on the matrix manifold. Interestingly, the dual parameters look like those in a tensor factorization: a (core) tensor, and individual mode-matrices. Experiments show some advantages of the proposed algorithms. [Related Works] Standard (l1) latent trace norm regularization [15] is related and it can be regarded as a baseline. [Strengths] I think this work is original and interesting. Contents are technically solid and informative. [Weaknesses] 1) Protocols of computational validation for some tasks are slightly unclear. For example, Authors did the training by 10% of full-samples and the testing by another 10% for image inpainting task. In normal sense, the validation should be performed by the other 90%. 2) Experimental results show some advantages of the proposed algorithms, however it is not very significant. Still competitive with state-of-art algorithms. 3) Another some validations such as noise robustness, outlier robustness, are also interesting to see and to improve the experimental parts. 4) Properties of l2 latent trace norm regularization are nice to discuss more. For example, difference from the Tucker based multi-linear low-rank decomposition is nice to discuss because Tucker decomposition (Rprecon) was slightly better than the proposed method for the movie recommendation task. Also, what kinds of tasks is the proposed method preferably applied for? I guess that the use of only proposed regularization is not the best for image/video inpainting task because another image priors (such as smoothness) will contribute to improve the performance. [Minor points] -- In Table 3, should T-svd for Ribeira 0.064 be also bold? --- after authors feedback --- Thank you for doing additional noise experiments.